# A Computational Interface to Translate Strategic Intent from Unstructured Language in a Low-Data Setting

**Pradyumna Tambwekar[1], Lakshita Dodeja[2],[*] Nathan Vaska[3],[*] Wei Xu[1], and Matthew Gombolay[1]**

[1]School of Interactive Computing, Georgia Institute of Technology
[2]Computer Science Department, Brown University
[3]Massachusetts Institute of Technology, Lincoln Laboratory

pradyumna.tambwekar@gatech.edu, lakshita_dodeja@brown.edu,
nathan.vaska@ll.mit.edu, {wei.xu, matthew.gombolay}@cc.gatech.edu

## Abstract

Many real-world tasks involve a mixed-initiative setup, wherein humans and AI systems collaboratively perform a task. While significant work has been conducted towards enabling humans to specify, through language, exactly how an agent should complete a task (i.e., low-level specification), prior work lacks on interpreting the high-level strategic intent of the human commanders. Parsing strategic intent from language will allow autonomous systems to independently operate according to the user's plan without frequent guidance or instruction. In this paper, we build a computational interface capable of translating unstructured language strategies into actionable intent in the form of goals and constraints. Leveraging a game environment, we collect a dataset of over 1000 examples, mapping language strategies to the corresponding goals and constraints, and show that our model, trained on this dataset, significantly outperforms human interpreters in inferring strategic intent (i.e., goals and constraints) from language (p < 0.05). Furthermore, we show that our model (125M parameters) significantly outperforms ChatGPT for this task (p < 0.05) in a low-data setting.

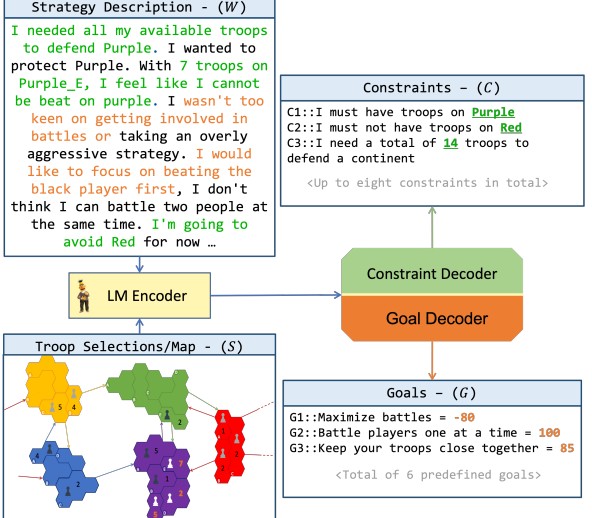

Figure 1: Our work aims to facilitate humans to specify their strategy to an AI system via language. Using the board game Risk as a simulated environment, we collect language descriptions of a strategy (top-left) corresponding to a player's troop deployments (bottom-left). The player's selections are shown by the white icons, and the grey and black icons denote the troops of the two opposing players. Each strategy corresponds to a set of goals (bottom-right) and constraints (top-right) The green and orange text corresponds to the language relating to constraints and goals respectively.

## 1   Introduction

Effective communication is essential for the proper functioning of organizational teams. "Commander's Intent" is a method for developing a theory of mind utilized in many domains such as the search and rescue, pandemic response, military, etc (Mercado et al., 2016; Rosen et al., 2002; Kruijff et al., 2014). Commanders and leaders often utilize the formulation of "Commander's Intent" to convey the tasks that need to be accomplished and engender an understanding of the criteria for success to their subordinates (Dempsey and Chavous, 2013). Commander's Intent could similarly function as

an effective scaffold to represent a human's strategic intent in a mixed-initiative interaction (Novick and Sutton, 1997). Commander's Intent provides a functionality for expert-specifiers to engender a degree of "shared-cognition," between an AI-collaborator and a human-specifier, by aligning the actions of the AI system to the human-specifiers values or reward function.

Commander's intent is formally represented by a set of *goals* and *constraints*. Goals (or preferences) are categorized as a desirable set of states or affairs that the agent intends to obtain (Moskowitz and Grant, 2009; Kruglanski, 1996) and constraints refer to conditions that are imposed on solutions

---

[*]These authors contributed to this paper while they were at Georgia Institute of Technology.

formulated by an agent (Nickles, 1978). Translating unstructured language-based strategy into this machine-readable specification is a non-trivial challenge. This translation could be conducted via a human interpreter, however, interpreters with the requisite expertise will not always be available. Alternatively, humans could utilize a structured interface to specify their intent. However, interfaces can become overly complicated, and humans become demotivated to work with an AI system when they cannot easily navigate the interface (Hayes, 1985). Enabling humans to express their strategic intent in everyday language provides an effective solution to these issues.

In this paper, we develop an approach to solve a task we call automatic strategy translation , wherein we learn to infer strategic intent, in the form of goals and constraints, from language. Prior work has developed methods to utilize language to specify policies of an AI agent (Tambwekar et al., 2021; Gopalan et al., 2018; Thomason et al., 2019; Blukis et al., 2019) or specify reward functions or tasks which can be optimized for, via reinforcement learning (RL) or a planner (Gopalan et al., 2018; Padmakumar et al., 2021; Silva et al., 2021a). However, our work is the first to translate language into goals and constraints, which can be applied towards constrained optimization approaches for directing agent behavior independent of the original human specifier. Unlike prior work, we focus on interpreting language description of complex gameplay strategies, rather than simple individual commands (e.g., "move from A to B; open the door").

First, we collect a dataset of over 1000 examples mapping language to goals and constraints, leveraging a game environment of Risk. Next, we fine-tuned a pretrained RoBERTa model (Liu et al., 2019), equipped with model augmentations and customized loss functions such as Order-Agnostic Cross Entropy (Du et al., 2021), to infer goals and constraints from language strategy specifications. Finally, we employ a human evaluation to test our approach. Recent work has shown that automated evaluation metrics for language models may provide a misleading measure of performance (Liang et al., 2022). Therefore, we design a head-to-head evaluation, whereby, we can directly compare our model to the average human intepreter. In addition to humans, we prompted ChatGPT to perform the same task on a held-out set of 30 examples. We computed the statistical difference between our

model and these baselines, providing a concrete measure of the relative efficacy of our approach. Our contributions are as follows:

- We propose one of the first complete machine learning pipelines including data collection, augmentation and model training for inferring structured strategic intent from human language.

- Through a human study, we show that our proposed approach can interpret goals and constraints from language descriptions better than the average human (p < 0.001).

- Through in-context learning, we evaluate Chat-GPT's performance to gauge the relative efficacy of our approach, and show that our approach significantly outperforms ChatGPT (p < 0.05).

## 2   Related Work

This section covers prior work on learning strategies from language, as well as methods and datasets to enable humans to specify AI-behavior in a mixed-initiative setting.

### 2.1   Learning strategies from Language

A common approach for specifying strategies through language has been through encoding language instructions, via planning-based representation languages, such as PDDL or LTL  (Williams et al., 2018; Bahdanau et al., 2018; Thomason et al., 2019; Tellex et al., 2020), or deep learning (Fu et al., 2019; Blukis et al., 2019; Gopalan et al., 2018). Such formulations facilitate the ability to constrain actions taken by the agent to the instruction specified, e.g. "Go around the tree to your left and place the ball." Another popular alternative is language-conditioned learning, where language is employed to specify a reward function, or a task (Silva et al., 2021a; Goyal et al., 2019; Andreas et al., 2017; Shridhar et al., 2022). Such approaches seek to improve the ability of an agent to complete a task(s) through intermediate language inputs, such as "take the ladder to your left". However, these approaches do not allow a supervisor to specify their strategic intent, such that the agent can complete it's primary task while still adhering to the specifier's plan. Recent work proposed a novel approach to mapping language to constraints and rewards via a dependency tree (Rankin et al., 2021), however their approach relies on a pre-trained grammar to extract a dependency tree, thus may not scale to human-like language.

Formally, the process of optimizing AI systems given goals and constraints has been broadly categorized as Seldonian Optimization (Thomas et al., 2019, 2017). In this framework, the goal is to optimize the priorities of an objective function while adhering to a given set of constraints as opposed to simply optimizing based on the reward or loss function. (Yang et al., 2020) proposed a Seldonian optimization approach to translate constraints into a feature representation, encoding invalid regions in the state space, which is then applied towards safe RL. However their application is restricted to learning to parse individual constraint statements such as "Don't get too close to the water," rather than facilitating constraint extraction from more realistic descriptions pertaining to an entire strategy. In our work, we provide a first-of-its-kind dataset, and correspondent model, to capacitate seldonian optimization through unstructured language.

## 2.2 Language and Strategy Datasets

Prior datasets for instruction following and policy specifications are often comprised of shorter instructions describing individual tasks. In contrast, our dataset consists of larger, unstructured descriptions of strategies which may be more reflective of potential strategy descriptions from in-the-wild users. Recent work has published a dataset of policy descriptions which are similar to the language descriptions we collect (Tambwekar et al., 2021) - however, they describe specific policies, rather than broad strategies for a task. Other datasets look to map language to trajectories or goals states within the trajectory (Padmakumar et al., 2021; Misra et al., 2018; Suhr et al., 2019). These datasets typically serve as a means of replacing physical demonstrations with language. These datasets lack explicit goals and constraints corresponding to the language collected, that can be applied towards seldonian optimization. Recent work provided a dataset with constraint statements (Yang et al., 2020) which are designer-specific; however, each constraint is associated with an isolated statement, making it unclear whether this approach will generalize to unprompted language describing multiple constraints. Unlike prior work, our dataset provides the ability to apply Seldonian optimization approaches from unstructured language. Furthermore, we conduct a study wherein we provide a human and ChatGPT baseline for our dataset to highlight the challenging nature of this task.

# 3 Natural Language Strategies in RISK

Our work aims to facilitate humans to specify their strategy or commander's intent to an AI system via language. In this section, we utilize the board game Risk to create a dataset that maps unstructured natural language descriptions of strategies to actionable intent in the form of goals and constraints.

## 3.1 Board Game – RISK

Risk (Gibson et al., 2010) is a multiplayer strategy board game of diplomacy, conflict, and conquest, which was first invented in 1957. The gameplay of Risk consists of four phases: Draft, Recruit, Attack, and Move. The draft phase is conducted at the start of the game wherein each player drafts an initial set of continents and deploys a fixed number of troops onto those continents. This allocation of troops is a crucial participatory task (Muller and Kuhn, 1993) which involves humans reasoning about their strategy and setting up for the rest of the game. Participants may choose any of the empty territories on the map to deploy their troops, with a wide range of strategies that may depend on their opponent's troop allocation. For example, a more conservative player may draft troops to only one continent for better defense, whereas a player with a more aggressive strategy may choose to spread out their troops. After the draft phase, each subsequent turn for a player involves iteratively conducting the recruit, attack, and move phases. Further details about Risk can be found in Appendix-I.

In our setting, we use a map layout that has 5 continents with a total of 21 territories/countries, as illustrated in Figure 1. Instead of real country names used in the Risk game, we use ad-hoc names for each continent (e.g., Red, Green, Blue, etc.) to mitigate participant bias. In the draft phase, each player takes turns to deploy 14 troops. The specific set of tasks that humans need to complete for our study include: (i) develop a strategy for Risk and deploy 14 troops after the two opposing players have completed their draft; (ii) provide six goals (on a 200-point scale) and up to eight constraints that were relevant to their allocation of troops and broader intents; (iii) use natural language to describe their overall strategy and the goals and constraints they considered. The troops of the opposing player are shown to the participants prior to completing these tasks. More details about this data collection process are discussed in Section 3.3.

## 3.2 Task Definition

Our goal is to develop a computational interface capable of inferring strategic intent from unstructured language descriptions of strategies. Formally, we define the task of Automatic Strategy Translation as follows: Given the troop deployments $S$, a map $M$, and the strategy $W$, which is a paragraph written in natural language, our task is to automatically derive a set of goals $G$ and constraints $C$. The troop selections $S$ include the name and number of troops for each territory drafted by the player. We have a total of 6 predefined goals, each of which takes a numeric value between $[-100, 100]$. This numeric value corresponds to whether the goal positively or negatively aligns with the strategy. For example, for the goal "maximize battles", 100 implies that the player intends to battle as much as possible, and -100 implies that the player intends to battle as infrequently as possible. Each constraint is comprised of a class and value. We restrict the number of possible constraints to 8 as a reasonable upper bound per strategy. To summarize, each example $\langle M, W, S, C, G \rangle \in \mathcal{D}$ consists of a strategy $W$ described in natural language, for a player's troop selections, $S$, on a map, $M$, from which $C$ and $G$ are the gold standard constraints and goals.

## 3.3 Data Collection

We collected a dataset $\mathcal{D}$ of 1053 unique examples by recruiting participants on Amazon Mechanical Turk and Profilic (pro, 2014). Firstly, to familiarize participants with the game, we designed a tutorial that provided a description and annotated examples to explain the rules of the game and the tasks that participants needed to perform. As a further measure of improving data quality, participants were quizzed on the rules of Risk to reinforce their understanding (full quiz has been provided in §A.2). They were given three attempts to answer correctly, after which they were shown the answers. Upon completing the quiz, participants began the task. We showed participants a map, which shows the drafted troops of the two opposing players, and asked them to provide their own troop deployments. Following their draft, participants are asked to provide the goals and constraints they considered for their gameplay strategy/deployments and finally provide a language description of their strategy. The language strategy they provided needed to have at least 200 characters. Each participant was asked to repeat this task 5 times to create 5 data points,

each time with a different map. The maps seen by participants were selected from a set of 15 unique initial troop settings.

Participants needed approximately 10 minutes per data point. Figure 1 depicts the format of our dataset. Our dataset included data from 230 participants. The average length of language descriptions in our dataset was 99.21 words, and the overall vocabulary size was 2,356 words. Additional details regarding our data collection protocol are available in Appendix A.

## 4 Automatic Strategy Translation

Following the data collection in Section 3, our goal is to leverage this dataset to develop a model that can perform the task of automatic strategy translation. Inferring strategic intent from language is a non-trivial endeavor as unstructured language can be vague thus leading to ambiguous interpretations. We seek to develop an approach capable of performing this task better than the average human, so as to enable strategy specification via language to reduce the potential risk of human errors or the need of third-party expert interpreters. In this section, we cover the technical details which make this task possible in a low-data setting.

### 4.1 Text Encoder

We adopted the pretrained RoBERTa model (Liu et al., 2019) as our encoder which is parameterized by $\theta$. The input sequence to our model is comprised of the language description of the strategy, $W = [w_1, w_2, \ldots w_{|W|}]$, and troop selections $S = [s_1, s_2 \ldots s_{|S|}]$, where each troop selection is comprised of the country name along with the number of troops placed on that country (e.g., $S = [Red\_A = 2, Red\_C = 8, Purple\_D = 4]$). The encoder learns the embedding function, which maps the text input, comprised of the strategy $W$ and selections $S$, to a $d$-dimensional real-valued vector which then be used towards predicting goals (§4.2) and constraints (§4.3).

Ordinarily, the final embedding for the single [CLS] token learned by RoBERTa, i.e., $E_\theta = BERT_{[CLS]}(W, S)$, is used for classification. In this work, we incorporate multiple classification tokens (Chang et al., 2023), each of which corresponds to an individual goal or constraint. For $i$th goal or constraint, we learn a separate classification embedding, $E_\theta^i = BERT_{[CLS_i]}(W, S)$. Using individual class-specific tokens improves the model

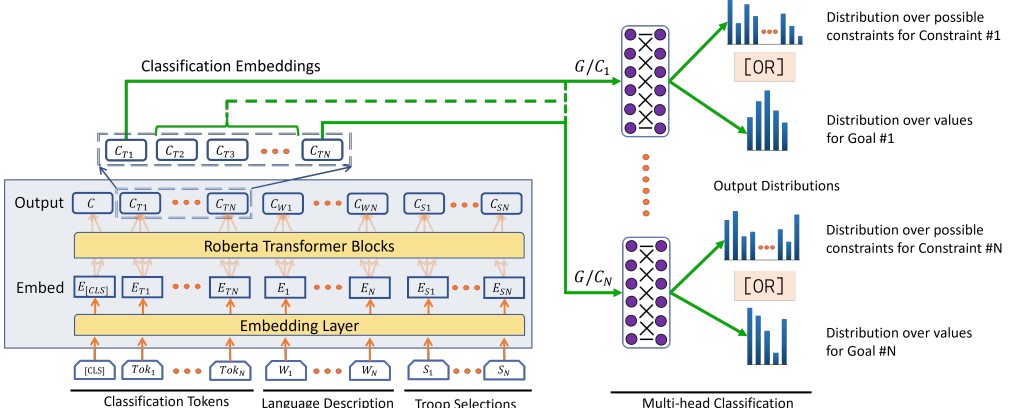

Figure 2: Illustration of our Automatic Strategy Translation model. The input to the model includes the classification tokens, language description, and troop selections (Section 4.1). The encoder then generates embeddings for each classification token, and passes them onto an individual classification head. Each classification head is a fully-connected layer that predicts a probability distribution for the respective goal (§4.2) or constraint (§4.3).

the capability to learn different attention weights corresponding to the classification embeddings for each goal or constraint. We utilize different encoders for predicting goals and constraints, which are parameterized by $\theta_g$ and $\theta_c$, respectively.

## 4.2 Goal Extraction Model

We treat the subtask of deriving goals from language as an ordinal classification task. Originally, in our dataset goals are specified as continuous values ranging from $[-100, 100]$, which we discretize by creating 5 uniform buckets, i.e., $[-100, -60)$, $[-60, -20)$, etc. That is, for each goal, we predict an assignment as a 5-class classification as:

$$P_j = L_{\phi_j}(E_{\theta_g}^j), \qquad (1)$$

where $P_j$ represents the probability distribution across assignments for $j$th goal and $E_{\theta_g}^j$ corresponds to the embedding from the encoder. Each goal uses a separate classification layer $L$ parameterized by $\phi_j$. The goal extraction model is trained on a dual-criteria loss function that combines cross-entropy (CE) and mean-square-error (MSE) loss:

$$\mathcal{L}_{goal} = \alpha\mathcal{L}_{CE} + (1-\alpha)\mathcal{L}_{MSE}, \qquad (2)$$

where $\alpha$ is a simple weighting hyperparameter. The addition of MSE loss helps to account for the ordinal nature of goal value predictions.

## 4.3 Constraint Extraction Model

Similar to the goal extraction model, the input to each classification head for constraint prediction is $E_{\theta_c}^k$, which corresponds to the classification embedding learned by the encoder for the $k^{th}$ constraint.

However, unlike for the goal extraction model, each of the eight constraint classification heads learns to predict the constraint itself rather than a value for a fixed goal. Therefore, the model needs to predict the set of unordered constraints $\{c_1, c_2, \dots c_8\}$, wherein each $c_k$ is predicted from the set of all possible constraints $C$ (190 total possible constraints). Each strategy can have a maximum of eight constraints, i.e., the set $C$ includes a null value.

While providing constraints during data-collection, participants merely assigned constraints to their strategy, but did not rank the ordering of constraints. As such, the order of constraints in our dataset does not necessarily correspond to the order in which each classification head needs to predict the constraints. Therefore, each classification head does not have a strict label it can utilize to compute a classification loss, making this task distinct from conventional sequence prediction or multi-class classification tasks. For instance, if the constraints predicted by the model are $\{C, \emptyset, B, D\}$ and the labels for this strategy are $\{A, B, C, \emptyset\}$, utilizing a standard classification loss function, such as cross-entropy, would result in a higher loss than what is representative of the prediction, as three out of four constraints have been predicted correctly. As such, this task requires a loss function that allows us to train our model to predict the correct constraints for a language strategy agnostic of the ordering of the labels. We chose to adopt a recently proposed loss function called Order-Agnostic Cross Entropy (OaXE) (Du et al., 2021). Intuitively, OaXE is defined as the cross entropy for the best possible alignment of output tokens.

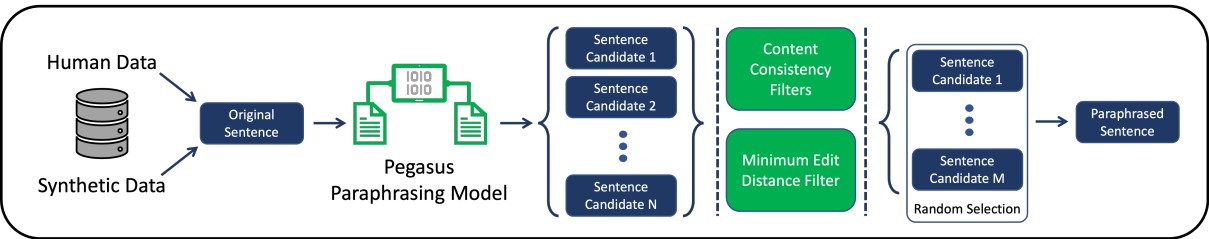

Figure 3: Pipeline for augmenting synthetic or human-created data (§4.4). A strategy description is first split into sentences, then passed into the PEGASUS (Zhang et al., 2020) paraphrasing model and data quality filter.

Let $O = \{O_1, O_2, \ldots O_{|O|}\}$ be the ordering space of all possible orderings of the target sequence of constraints, where each $O_l$ is one possible ordering of the target tokens. The final loss function is computed as:

$$\mathcal{L}_{OaXE} = -log P(O^*|X) \qquad (3)$$

where $O^*$ represents the best possible alignment from $O$. This alignment is computed by applying the Hungarian algorithm, after casting this problem as maximum bipartite matching (Du et al., 2021). As our final loss function, we follow Du et al. (2021) in combining OaXE with cross-entropy loss:

$$\mathcal{L}_{constraint} = T_m * \mathcal{L}_{CE} + (1 - T_m) * \mathcal{L}_{OaXE} \qquad (4)$$

where $T_m$ is a temperature parameter that is logistically annealed from 1 to 0. In our case, cross entropy ($\mathcal{L}_{CE}$) is computed using the default ordering of labels in our dataset.

### 4.4 Data Augmentation Methods

Finally, we applied data augmentation procedures to improve our model's performance. First, we randomly generated 4000 unique sets of goals and constraints, and applied a text template to produce descriptions to develop a Synthetic (S) training corpus. For example, the constraint, "I must have troops on Red" could be represented as "My strategy is to take over Red," or "I need a large army on Red," or "I need to place troops on Red." We further augmented this synthetic corpus with a pretrained PEGASUS (Zhang et al., 2020) paraphrasing model to create an Augmented-Synthetic (AS) dataset. We split each language description from the synthetic corpus into individual sentences and employed the paraphrasing model to generate candidate paraphrases. Sentences that replaced important keywords, such as continent names, or were too similar to the original sentence in terms of edit distance were removed. We randomly chose

a sentence from the remaining candidates as a replacement sentence, and combined the replacement sentences to form an augmented data point (see Figure 3). The two Synthetic datasets (S, AS) were used to pretrain our model prior to training on human data. The same techniques were also applied to our human dataset to form a Augmented-Human dataset (AH). Our final Augmented-Human data set is a version of our original crowdsourced dataset where each example is rephrased using our augmentation pipeline and is twice the size of our original human dataset. We experiment with utilizing the AH dataset in place of the original human dataset to see if the added diversity in our corpus through paraphrasing improves downstream performance. Examples of Synthetic (S), Augmented-Synthetic (AS), and Augmented-Human (AH) data are provided in Figure 6 in the Appendix.

## 5 Experiments

This section will present the empirical evaluations of our approach. We design two evaluation experiments to contrast our model's performance with humans, as well as ChatGPT trained to perform our task through in-context learning. Both human and ChatGPT performance was computed using the 30 held-out examples in our test set. We statistically measure the difference in the average number of goals/constraints predicted correctly per data point between our model and the two baselines (Human + ChatGPT). We conclude with an ablation analysis across the model and data augmentations utilized in this approach.

### 5.1 Human Performance

In our first study, we ask how well the average human can perform on the task of parsing strategic intent from language (see Table 1). We recruited 114 participants for our study from Prolific. Participants begin with a tutorial of the task and are provided an annotated example explaining how to

| Baseline | Goals (Total = 6) | Constraints (Total = 8) |
|---|---|---|
| Model (Ours) | **2.76 $\pm$ 1.05** | **5.53 $\pm$ 1.26** |
| Human | 1.87 $\pm$ 1.12 | 4.28 $\pm$ 1.83 |
| ChatGPT | 2.10 $\pm$ 1.27 | 3.80 $\pm$ 1.51 |

Table 1: Mean and standard deviations for the number of correct predictions of each approach.

assign goals and constraints given a language description and map. Following this tutorial, each participant is provided three randomly selected maps and language descriptions from our test set of 30 unique data points and is asked to annotate the goals and constraints for each given strategy. Our study included attention checks to ensure participants who were submitting random responses could be excluded. The average time taken for our study was 21 minutes, and participants were paid $3.6 for completing our task. We utilized a data filtering rubric to identify and remove individual data points which were inadequate or were from participants who appeared to blatantly ignore or misunderstand the instructions. The rubric is included in Appendix F. After filtering, a total of 270 responses remained.

## 5.2 ChatGPT Performance

We also evaluate ChatGPT (GPT-3.5 Default) as a baseline for our task (see Table 1). We design a 1000-word language prompt to train ChatGPT to perform the same task (see full prompt in Appendix G.1). This prompt includes a description of the environment and task, as well as an annotated example translating goals and constraints from language. Crucially, we design our prompt such that ChatGPT receives the same information that humans receive in our study in §5.1. Following this prompt, we iteratively input each strategy and troop deployment in our test set and store the constraints selected by ChatGPT. The additional prompt engineering we conduct is to notify ChatGPT when it makes formational mistakes while predicting constraints, such as predicting more than the maximum number of constraints or creating new constraint classes.

## 5.3 Results for Goal Extraction

The average number of goals predicted correctly per map can be seen in the first column of Table 1. We applied multivariate linear regression to compare the resuls of our model with our ChatGPT and human baselines, with Akaike information criterion (AIC) as our Occam's razor. AIC is a mathematical

| Model Type | Data | Pretraining | Accuracy (Std) |
|---|---|---|---|
| RoBERTa$_{base}$ | – | – | 44.37 (1.33) |
| **w/ troop** | **AH** | **AS** | **46.04 (1.85)** |
| w/ troop + [CLS$_i$] | AH | AS | 45.52 (1.48) |
| w/ troop + [CLS$_i$] | AH | S | 45.32 (1.01) |
| w/ troop + [CLS$_i$] | AH | – | 45.89 (1.26) |
| w/ [CLS$_i$] | AH | AS | 44.29 (1.14) |
| w/ troop + [CLS$_i$] | H | – | 45.07 (1.33) |

Table 2: Ablation study (10-fold cross-validation) with respect to model and data augmentations for **goal extraction**. H: the human-created dataset (§3.3); S: the synthetic dataset created from templates; AH/AS: the augmented version of H/S via paraphrasing (§4.4). [CLS$_i$] represents the use of individual classification tokens for each goal/constraint (§4.1); "troop" represents the inclusion of troop selections as a part of the input.

| Model | Data | Pretraining | Accuracy (Std) |
|---|---|---|---|
| RoBERTa$_{base}$ | H | – | 62.60 (1.60) |
| **w/ troop + [CLS$_i$]** | **H** | **S** | **68.21 (1.08)** |
| w/ troop + [CLS$_i$] | AH | S | 67.79 (1.58) |
| w/ troop + [CLS$_i$] | H | AS | 67.09 (1.28) |
| w/ troop | H | S | 65.96 (1.12) |
| w/ troop + [CLS$_i$] | H | – | 65.76 (1.13) |
| w/ troop + [CLS$_i$] | AH | – | 65.52 (1.42) |
| w/ [CLS$_i$] | H | S | 65.31 (1.12) |

Table 3: Ablation study (10-fold cross-validation) for **constraint extraction**.

method for determining a model-fit so as to choose the regression model which best fits our data. For the goals model, we modeled each baseline (human vs. model vs. ChatGPT) as a fixed effects co-variate, and the datapoint number as a mixed effects variable. The datapoint corresponded to the numerical index (between 1 - 30) of the datapoint from the test set. We performed the Levene's test (Glass, 1966) to show homoscedasticity ($F(2, 327) = 0.5435$, $p = 0.581$). The residuals for our model were not normally distributed; however, prior work has shown that an F-test is robust to non-normality (Blanca Mena et al., 2017; Cochran, 1947). Therefore, we proceeded with our linear regression analysis. The dependent variable within our analysis was the number of goals predicted correctly. An ANOVA with respect to our dependent variable yielded a significant difference across conditions ($F(2, 299.95) = 10.605$, $p < 0.001$). A Tukey post-hoc test (Abdi and Williams, 2010) for pairwise significance further revealed a significant difference between the performance of our model vs humans ($p < 0.001$) and vs ChatGPT ($p < 0.05$), i.e., our approach was able to significantly predict

goals better than humans and ChatGPT.

## 5.4  Results for Constraint Extraction

The average number of constraints predicted correctly per map can be seen in column 2 of Table 1. To compare our constraint prediction model, to our human and ChatGPT baselines, we conducted a non-parameteric Friedman's test (Pereira et al., 2015). We could not employ a multivariate regression analysis, as the regression model for constraints did not satisfy the assumption of homoscedasticity as per Levene's test ($F(2, 327) = 5.4294, p < 0.01$). The Friedman's test yielded a significant difference across conditions for the task of predicting constraints ($\chi^2(2, 90) = 16.768, p < 0.001$). A further pairwise Wilcoxon signed rank test (Woolson, 2007) revealed a significant difference between humans and our model ($p < 0.001$) as well as ChatGPT and our model ($p < 0.001$), indicating that our approach is not just able to significantly outperform humans, but also ChatGPT for inferring constraints from language.

## 5.5  Discussion

Our results emphasize that inferring strategic intent from language is a non-trivial task, as language interpretation can be subjective and malleable. Chat-GPT is capable of performing novel tasks such as text classification (Li et al., 2023), mathematical problem solving (Frieder et al., 2023), and information extraction (He et al., 2023). through in-context learning. However, despite these capabilities, our model was found to significantly outperform chatGPT for inferring strategic intent from language. Success in highly specific and complex language interpretation tasks, such as ours, requires the model to build an understanding of the domain and the task itself as generic language interpretation learned by the majority of pretrained language models may not be applicable.

Recent work on evaluating open question-answering on a challenge-dataset has shown that even for large-scale language models with between 6B-100B parameters, none of these models outperformed humans (Peinl and Wirth, 2023). By developing a computational interface which can infer strategic intent from language significantly better than humans, we show the usefulness of our pipeline towards solving complex domain-specific task in a low-data, -resource setting.

| Baseline | Constraints | Goals |
|---|---|---|
| Roberta-base (Best) | **68.21 (1.08)** | 46.04 (1.85) |
| GPT-Neo 125M (Best) | 65.22 (1.21) | **46.08 (0.73)** |

Table 4: This table depicts the performance when the roberta-base encoder is substituted with a SOTA autoregressive model, i.e. GPT-Neo (125 million parameters).

## 5.6  Ablation Study

Tables 3 and 2 provide the results from ablating each model augmentation discussed in Section 4. The effects of these augmentations are more prominent in the model for predicting constraints ($\sim 6\%$ performance boost) than predicting goals ($\sim 1.5\%$ performance boost). For the constraints model, when any parameter, i.e. troop selections, pretraining, or CLS-Tokens, were removed, the accuracy dropped by $\sim 3\%$ individually. For predicting goals, the inclusion of troop selections was the only model-augmentation which seemed to have a decisive impact performance, as all models with selections had an accuracy of $\sim 1\%$ more than those without. We attribute the difficulty in improving the performance of the goals model to the contextual ambiguity for values assigned to each goal. Participants may not always follow the same metric while specifying goal values. Each participant could have a unique interpretation, for what any rating between -100 to 100 means for a particular goal, and description of that value through language(see Appendix for the data distribution corresponding to each goal). This disparity in interpreting values could be affecting the consistency of language descriptions for goals in our dataset.

Finally, the last ablation conducted studied the effect of the type of encoder utilized in our approach. Therefore, we performed a comparison with a model which replaced the encoder with a SOTA pretrained autoregressive model. We utilized GPT-Neo (Black et al., 2021) for our experiments, as it has the same number of parameters as Roberta-base (125 million). Our findings (see Table 4) show that utilizing an autoregressive model as our encoder offers no benefits to a roberta-base model, the GPT-Neo model performed equivalently for predicting goals and about 3% worse for the constraints model.

## 6  Conclusion

In this paper, we develop a novel computational interface to automate inferring strategic intent, in the

form of goals and constraints, from unstructured language descriptions of strategies. We develop a new benchmark for our dataset and broader task, and further conduct a novel head-to-head evaluation to determine the relative efficacy of our approach. We show that in a low-data setting, our approach towards inferring goals and constraints from language strategy descriptions can significantly outperform humans for the same tasks. Furthermore, we also found that our approach, with only 125 million parameters, was able to significantly outperform ChatGPT for inferring strategic intent from language. Our work endows researchers with valuable tools to further seldonian optimization approaches for mixed-initiative interaction.

## Future Work

To measure ChatGPT performance, we employ a one-shot chain-of-thought prompt method with a detailed instructions of the task. We chose this method to maintain consistency between the information shown to humans and ChatGPT. Future work may explore ablations on the size of the initial prompt or the number of annotated examples in the prompt to tune the performance of ChatGPT on our strategy translation task. Secondly, an important next step that stems from this research pertains to multi-round inference and updating the initially learned strategy. In future work, it would be helpful to develop methods to allow users to modify their initial strategy throughout the game or task as their goals or values change. These methods could utilize approaches proposed in prior work wherein language inputs were leveraged to change the sub-goals that an agent is considering (Fu et al., 2019; Goyal et al., 2019). Furthermore, recent work has shown promise for the capabilities of ChatGPT/GPT-3.5 towards dialog-state tracking and task-oriented dialog (Labruna et al., 2023; Heck et al., 2023). Future work could also formulate this task of updating the initial strategy over the course of the game as a goal-oriented dialog, and tune GPT-3.5 or GPT-4 to update a user's initially translated strategy after multiple rounds of the game through language feedback.

## Limitations

Firstly, we asked participants to provide natural language descriptions after providing their structured intent in the form of goals and constraints. This potentially biased the participant towards specifically referencing the terminology utilized in the goals and constraints. While our dataset provides explanations that are the closest to natural, human-like descriptions of strategies, an important next step would entail comparing how our model performs on strategies collected "in-the-wild." Secondly, in this paper we assume that utilizing language is more accessible than learning to use mathematical specifications directly to specify their intent to an intelligent agent. However, we do not test whether this assumption bears out in practice. In future work, we hope to develop a human-subjects study to confirm this hypothesis. Finaly, despite converting language to goals and constraints, in this work we do not directly train a seldonian optimization approach. In this work, we focus on showing the capability of our machine learning pipeline in a low-data setting. However, we have provided all the components needed to train a reinforcement learning approach for an RL-agents constraining behavior through unstructured language (including a novel open-AI RL domain for the game Risk, see Appendix). Developing this approach is currently outside the scope of this work, and we thereby leave this exploration for future work.

## Ethics Statement

As pretrained large-language models are utilized in our approach for automated strategy translation, we need to be cognizant of the prevalence of bias within these models. If these systems are translating strategies in safety-critical settings, it is important to make sure that the language models make the decisions solely based on the provided context rather than any inherent bias. Many sets prior work have studied approaches to identify and mitigate bias (Abid et al., 2021; Silva et al., 2021b; Guo et al., 2022; Viswanath and Zhang, 2023). We encourage authors to seek out such works prior to deploying any strategy translation module, towards a real-world task.

## Acknowledgements

This work was supported by the Office of Naval Research under awards, N00014-19-1-2076, N00014-22-1-2834, N00014-23-1-2887, and the National Science Foundation under award, FMRG-2229260. We also thank Konica Minolta for their contribution to this work via a gift to the Georgia Tech Research Foundation.

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

## A  Additonal Data Collection Details

Our study applied participatory design principles (Muller and Kuhn, 1993), to ensure that participants were engaged in the task and provided meaningful strategy descriptions. Each participant was initially given a partially setup map, where two other "opponents" had placed their troops. The participant was then asked to provide their troop placements, based on these initial placements. In Risk, the initial troop placements have a substantial impact on the strategies that a player can pursue for the rest of the game. As such, troop initialization provides a stand-in for a player's overall strategy in a game. By asking participants to participate in an actual aspect of the gameplay, e.g. deploying troops, participants were encouraged envision future situations and think about how their decisions could affect future gameplay and develop grounded strategies that could actually function as viable Risk gameplay strategies.

Next, participants were asked to provide the goals and constraints which they considered after selecting their troop placements. These specific goals and constraints were selected as they cater to potential strategies that could be employed while playing Risk. The presence of these templates provided a scaffold within which participants, who may or may not have any experience with Risk, could ground their strategies. However, it is important to acknowledge the presence of an inductive bias, due to the specific wording of the goals and constraint templates, which could have impacted the strategies submitted by the participants. For goals, participants were asked to rate how important each goal was to their strategy on a scale of -100 to 100. A score of -100 indicated that pursuing the goal was completely detrimental to their strategy, while 100 indicated that pursuing the goal was essential to their strategy. For constraints, participants were provided 9 constraint templates, and were asked to select and fill in the appropriate constraint that was represented in their strategy. Participants were required to provide at least three constraints to ensure that they did not skip this question. The specific goals and constraints in our dataset can be depicted in Table 5. Finally, participants were asked to summarize their strategy for the given map as a language description. Participants were encouraged to include references to their goals and constraints, but these descriptions were otherwise unprompted. Participants were paid up to $8.5 based on the number of adequate responses submitted. The payment scale was updated if the average time taken significantly changed.

As mentioned in the paper, we created three additional augmented datasets from our original corpus. Figure 6 provides some examples of the effect of the various augmentations we employed in each augmented dataset. Our full dataset can be found at the following anonymized Github repository - Anonymized Data Repository .

### A.1  Data Cleaning/Filtering

We performed the least possible modifications to participant's responses to ensure responses were self-consistent while preserving the integrity of the organic data collection task. If a participant specifically referenced a goal or a constraint in their language, and did not include it in their response, then their response was modified to include it, and vice versa. We also corrected typos within a participants specifications, such as if they meant to reference the "Blue" continent instead of the "Red" continent. If a response was not salvageable without minimum modifications, the response was thrown out. Discarded responses included responses where participants simply did not understand the task or submitted blatantly insincere responses such as copying text from the study multiple times to reach the character limit. These decisions were made upon agreement of multiple reviewers.

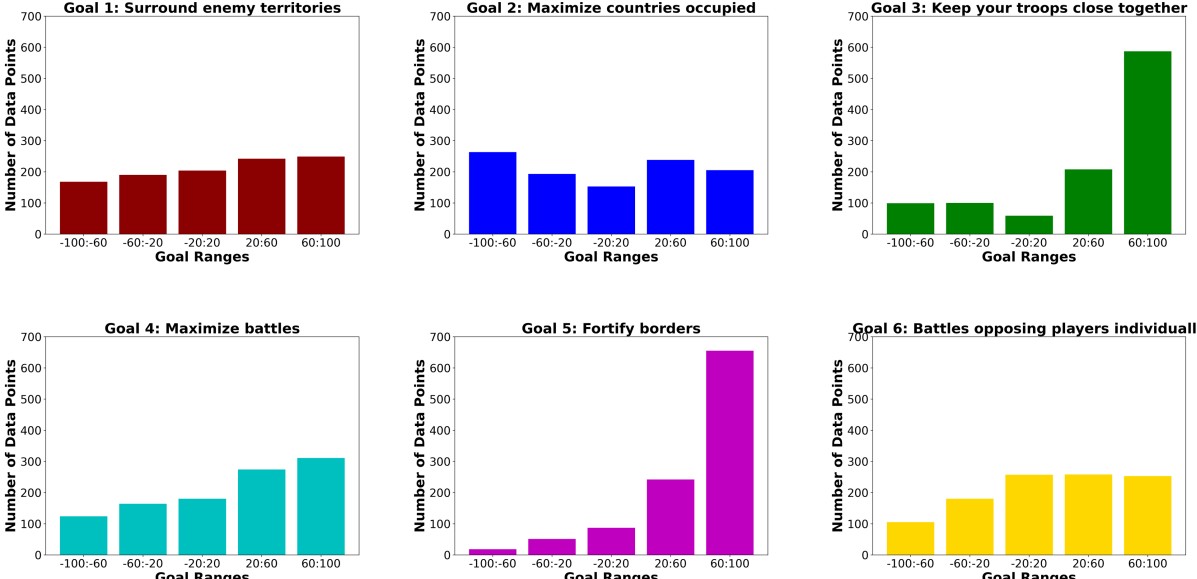

Figure 4: Distribution of assigned values for each goal. The titles for each goal have been shortened for readability.

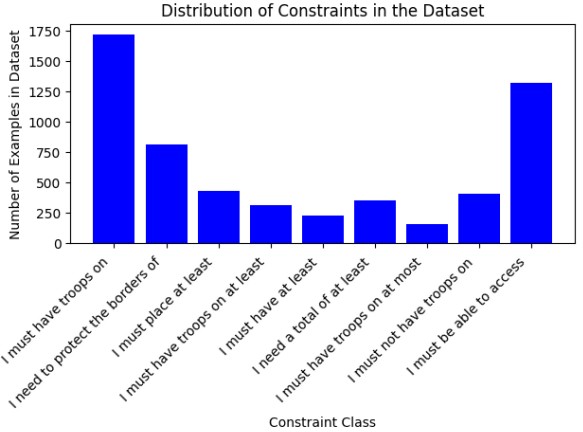

Figure 5: Distribution of assigned values for each constraint type

## A.2 Data Collection Quiz

In order to ensure that participants understood the rules of Risk prior to providing strategies for our dataset, each participant was asked answer a five question quiz. Participants needed to answer all questions correctly to proceed. Participants were given three tries to answer the questions after which they were shown the correct answers. The five questions in our quiz were as follows (correct answers to each question are in bold):

1. Which of these are **NOT** a phase in the game?

    (a) Attack
    (b) Recruit
    (c) **Control opponent's troops**
    (d) Maneuver

2. What is the objective of the game?

    (a) Control the rightmost continent
    (b) Have the maximum number of island territories
    (c) Have the most territories after 10 turns
    (d) **Occupy all territories on the board**

3. Which of these decides how many troops you receive at the start of each turn? (TWO CORRECT ANSWERS)

    (a) **The number of territories you control**
    (b) The number of coastal territories on the map
    (c) They physical size of the board game
    (d) **The number of continents you fully occupy**

4. Which of the following statements are correct about attacking enemy territories in the game? (TWO CORRECT ANSWERS)

    (a) When you attack a territory you've already attacked, your attack points are doubled

    (b) **You CANNOT attack in the opposite direction of the arrows**

    (c) **You can only attack territories you have access to**

    (d) You can never attack a territory in the same continent

5. Which of the following statements are true regarding how attacks are conducted? (TWO CORRECT ANSWERS)

    (a) A player with scattered troops always wins

    (b) A player attacking from the left side always wins

    (c) **Both players roll a number of dice dependent on the number of their troops involved in the battle to decide the outcome**

    (d) **A player can attack with up to 3 troops and defend with up to 2 troops in one battle**

## B  Dataset Utility

This section provides a brief discussion on the potential future utility of our collated dataset. Firstly, this dataset provides strategy specifications in Risk that can be used to test seldonian optimization approaches in future work. Our dataset provides the first such instance language descriptions of strategic intent. Future work can analyze the flaws and strengths of our data to modify our data collection protocol and generate the specific examples they may need for their individual applications. However, there are many tangential applications for this data that are unrelated to the use-case specified in this paper. There is a dearth of natural language datasets which contain language with human-like speech patterns that is not scraped from internet-corpora. Many NLP techniques can be applied to further study this language data such as summarization, to figure out whether these policies can be summarized into a more easily digestible format, sentiment analysis, for broadly categorizing the language description into *aggressive*, *defensive*, etc,

or Q&A comprehension-based methods, to train AI agents to answer questions regarding a user's preferences by reading their strategy description.

## C  Dataset Distributions

The data distribution for goals and constraints selected by participants are shown in Figure 4 and Figure 5 respectively. For Goals 3 (Keep your troops close together) and 5 (Maximize Battles) participants tended to skew towards answers in the 60-100 range. For the other goals, the responses were relatively uniform. On average, participants submitted 5.62 unique constraints per response.

## D  Implementation Details

Hyperparameters for both models were computed through a grid search over parameters. The constraints model was trained for 10 epochs with a batch size of 16 using a learning rate of 0.0005. The goals model was trained for 25 epochs with a batch size of 8 using a learning rate of 0.00001. The constraints model was Both models utilized an AdamW optimizer. The constraints model employed a cosine learning rate scheduler, and the goals model employed a linear learning rate scheduler. We hold-out 30 randomly selected examples for our human/ChatGPT evaluation (Section 5). We split the remaining 1023 examples into a 85/15 train/validation split to perform our grid search over hyperparameters. Finally, to report the accuracy of our model we computed the 10-fold cross-validation accuracy on the best performing hyperparameter setting. The best performing model for predicting constraints was pretrained on the synthetic corpus and trained on the un-augmented human corpus. The best goals model was pretrained on the synthetic-augmented dataset and trained on the human-augmented dataset. All experiments were conducted on a 48GB NVIDIA Quadro RTX GPU. Our code can be found at the following anonymized repository for further reference - Anonymized Code Repository.

## E  Human Evaluation Study - Additional Details

In this section, we report some additional details regarding our human-evaluation experiment. Firstly, we report that on average, the difference between scores for a participant's first and last response was -0.2143 for goals and -0.0102 for constraints, indicating that there is a negligible impact of factors

| Synthetic Data | Synthetic-Augmented Data |
|---|---|
| Why would I care about battling. I plan to attack players in the game one at a time. I don't think I can handle having troops on more than 2 continents. I need to spread my troops out as far as possible. I can't win if I put any troops on Blue. I need to place troops on at least 5 countries. This time I will use a different strategy. I need to have troops on at least 5 continents. I don't intend to control continents. | I don't know why I care about fighting. I plan to attack players in the game one at a time. I don't think I can handle having troops on more than 2 continents. My troops need to be spread out as much as possible. If I put any troops on Blue, I will not win. I need to place troops on at least 5 countries. I will be using a different strategy this time. I need to have troops on at least 5 continents. I don't intend to control continents. |
| **Human Data** | **Human-Augmented Data** |
| I am going to attack and take over green c. That country is ripe for the taking since I have cut it off from other grey troops. I also want 4 troops to present a strong force in green a in case of a grey attack from yellow d. Once the green continent is secure I will look to move my armies out to the red continent to battle black there. Hopefully, while this is going on grey and black will be fighting over yellow and blue, but in case they don't I'm keeping all of my troops together on Green | I am going to attack and take over green c. Since I cut it off from other grey troops, that country is ripe for taking. I also want 4 troops to present a strong force in green a in case of a grey attack from yellow d. I will move my armies to the red continent to fight black once the green continent is secure. Hopefully, while this is going on grey and black will be fighting over yellow and blue, but in case they don't I'm keeping all of my troops together on Green. |

Figure 6: Examples of data from Synthetic (top-left), Synthetic-Augmented (top-right), Human (bottom-left) and Human-Augmented (bottom-right). Highlighted sections represent the specific sentences changed by our augmentation procedure.

| Goals | Constraints |
|---|---|
| G1 : Surround enemy territories | C1 : I must have troops on (**continent**) |
| G2 : Maximize number of countries occupied | C2 : I must not have troops on (**continent**) |
| G3 : Keep our troops close together | C3 : I must be able to access (**continent**) in one move |
| G4 : Maximize battles throughout the game | C4 : I need to protect the borders of (**continent**) |
| G5 : Fortify borders for the continents you control | C5 : I need a total of at least (**number**) troops to defend a continent |
| G6 : Battle opposing players one at a time | C6 : I must have at least (**number**) countries |
| | C7 : I must have troops on at least (**number**) continents |
| | C8 : I must place at least (**number**) troops to effectively defend a country |
| | C9 : I must have troops on at most (**number**) continents |

Table 5: Goals and Constraints Selected for our Dataset

such as cognitive load or a learning curve. Secondly, it is important to note that we did not have the same number of responses per map from humans, as the map condition was randomly assigned to each participant. While this may slightly impact the results of the constraints model, as we aggregated performance across maps, due to the strong significant difference across baselines, it is unlikely to change our result.

## F  Human Evaluation Study - Data Filtering Rubric

Next, we cover the rubric we applied to filter data for the human-subjects study. Each response was independently evaluated by two graders and was included if both graders deemed it acceptable as per the predefined rubric. The rubric was as follows:

1. If constraints clearly don't match the selections for locations or access
   - e.g. if someone has selected, "I must have troops on Blue" when there are no troops on Blue

2. If someone has submitted invalid constraints
   - e.g. If someone selects both "I need troops on at least 2 continents" + "I need troops on at most 1 continent"
   - If someone mistakes "country" for "continent"

3. If someone has selected the same value for all goals (or values within a small range, say +-10), when this clearly does not align with the strategy
   - e.g. someone selects –100 for all goals when the strategy involves protecting a continent

## G  ChatGPT Prompt

We utilized the following prompt for ChatGPT which included a description of the domain and task, as well as an annotated example.

### G.1  Full Prompt

Reading the following section carefully will provide you with the information needed to complete

this task.

Risk is a board game in which an army commander tries to take over the world by defeating all enemy troops and controlling all countries. Risk is a simplified version of real conflict, and has rules designed to reflect this. These include the following:

- Players control countries by having troops in them

- The more countries and continents a player controls, the more resources they get

- Players win countries from other players by battling with their troops

- The more troops a player has when battling, the more likely they are to win

- Players can only attack or be attacked by countries that are next to them

In this task, you will be asked to provide a set of constraints corresponding to the human player's strategy for the board game Risk. This includes their troop placements and a text description, which explains why the player decided to place their troops and how they plan to win this game of Risk given their opponents' choices.

Your task will be to think about the player's strategy (selections and description) and predict what their constraints are with respect to the strategy. Constraints are rules that you think need to be followed to successfully execute a strategy.

**CONSTRAINTS:** *Note: For predicting goals, this section would be replaced with a description of what goals are*

Constraints are comprised of constraint classes and constraint values. Your job is to assign constraints to the human's strategy. Each constraint is comprised of a constraint class and a constraint value. You will be provided a list of possible constraint classes and values to choose from. You may choose the same class of constraint more than once, but you may not submit duplicate constraints. For example, you may submit "I must have troops on Green" and "I must have troops on Blue" but you may not submit "I must have troops on Green" twice. Choose all constraints relevant to the strategy. You may choose up to 8 constraints per strategy.

The constraints you can choose from are

- I must have troops on [Continent]

- I must not have troops on [Continent]

- I must be able to access [Continent] with one move

- I need to protect the borders of [Continent]

- I need a total of at least [Number] troops to defend a continent

- I must have at least at least [Number] countries

- I must have troops on at least [Number] continents

- I must place at least [Number] troops to effectively defend a country

- I must have troops on at most [Number] continents

The possible constraint values you can choose from are

- Continent - Blue, Green, Yellow, Red, Purple

- Number - 1,2,3,4,5,6,7,8,9,10,11,12,13,14

Our modified RISK Map contains 5 continents - Red, Green, Purple, Yellow and Blue. Each continent is made up of countries. Red continent has 3 countries, Green has 5 countries, Purple has 5 countries, Yellow has 4 countries and Blue has 4 countries. Green_A, Yellow_B, Blue_C, etc. are referred to as countries or territories Green, Yellow, Blue, Red, Purple are referred to as continents. Continents also have different connections between them through which the troops can move. These connections are one way i.e troops from the source country can only move to the destination country and not the other way round.

The map has the following connections - Yellow_D is connected to Green_A, Greed_D is connected to Red_A, Red_A is connected to Green_D, Red_B is connected to Purple_E, Red_C is connected to Yellow_B, Red_C is connected to Blue_B, Blue_A is connected to Yellow_C, Yellow_C is connected to Blue_D, Blue_C is connected to Purple_A, Purple_A is connected to Green_E and Green_E is connected to Purple_A

We will now give you a tutorial on how to ascertain the goals from a human player's strategy and placements on the RISK board.

The two opposing players are denoted by the "grey" and "black" player. In this scenario, the grey player has placed its troops on the following territories - 5 troops on Yellow_C, 4 troops on Yellow_D, 1 troop on Red_A, 2 troops on Red_B, 2 troops on Red_C. The black player has placed its troops on the following territories - 4 troops on Blue_A, 2 troops on Blue_C, 2 troops on Green_E, 5 troops on Purple_A and 1 troop on Purple_B.

Now that you have seen where the opposition troops are, you will now be shown how the human player has decided to deploy their troops and the strategy they used.

The human player (white) has placed 14 troops to battle the opponents. They have placed the troops on the following territories - 7 troops on Purple_E, 5 troops on Purple_C and 2 troops on Purple_D. You will now be guessing the constraints the human player (white) focused on while coming up with their strategy. The following text contains the human player's description of the strategy they used to place their troops. It is critical that you read this description, as it contains information about the constraints considered by the human player.

"I put all my troops in Purple, because I felt as though I needed all my available troops to defend Purple. I wanted to protect Purple. With 7 troops on Purple_E, I feel like I cannot be beat on purple. I wasn't too keen on getting involved in battles, or taking an overly aggressive strategy. I would like to focus on beating the black player first, I don't think I can battle two people at the same time. I'm going to avoid Red for now since it seems to be the hardest continent to control. "

We will now show you how to determine constraints from a strategy and via an example. Please carefully review the example and use the given information about both selections and text to fill out constraints for this strategy.

An appropriate set of constraints for the strategy shown above would be

- I must have troops on Purple

  – Reason: The player mentioned that "they put all their troops on Purple"

- I must not have troops on Red

  – Reason: The player mentioned that "they would like to avoid Red for now"

- I must place at least 7 troops to effectively defend a country

  – Reason: The player mentioned that "with 7 troops on Purple_E, I cannot be beaten on Purple"

## H  Risk Reinforcement Learning Simulator

We have shown that our proposed computational interface can remove the need for human-interpreters for the task of parsing intent from unstructured language. However, to test how well commander's intent interpreted from language can be applied towards optimizing an agent's behavior, we require a reinforcement learning domain to train our agent. As such, to enable seldonian optimization, via unstructured language descriptions, we developed a novel open-ai gym environment for simulating Risk gameplay. This environment closes the loop on the methods presented in this paper by providing all the necessary components for humans to specify their intent to an AI agent and evaluate whether their specifications have been satisfied by the learnt agent. Our environment also provides an additional means of collecting data and conducting studies for human-specification within multi-player team scenarios.

For this task, we adapted an existing open-ai gym environment for Risk (Andeol, 2018). We modified the codebase to allow for RL agents to be trained to play all phases of Risk, according to the setup utilized in our approach. We also developed a pygame-UI for our simulator (see Figure 7). A detailed description of the functionality of the domain and the state space is provided in the appendix. In future work, we aim to leverage our domain to develop approaches which allow humans to constrain an agent's optimization methods through human-like language specifications of intent, which has not been accomplished in any prior work. We also provide a link to an anonymized github repository with the risk environment for further reference - Anonymized Gym-Risk Environment

## I  Risk Domain - Additional Domain Information

This section provides additional information about our setup for Risk Domain. In our version of Risk, the ego player (Alpha), plays against two opponents (Charlie and Bravo) whose actions are controlled by a pre-determined heuristic. The gameplay within our Risk simulator is comprised of four phases

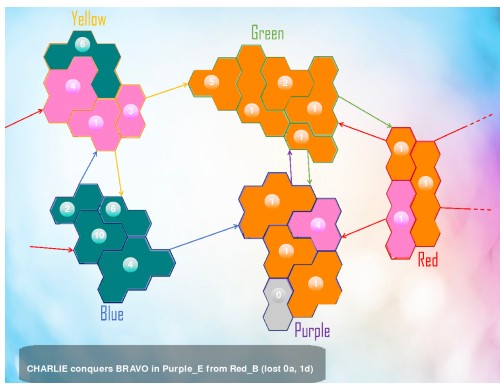

Figure 7: This figure shows our Risk simulator with the playable (teal) and two other (orange and pink) agents.

1. Drafting - Players draft their initial troops on empty territories.

2. Reinforce - Players assign reinforcements to their existing territories.

3. Attack - Players can choose to attack a neighboring territory with their troops.

4. Freemove - Players can move their troops between their territories.

The game begins with a drafting phase. During this phase, the agent decides where to place their initial 14 troops amongst the available territories. The two opposing players draft their troops before the agent is allowed to draft any troops. The opposing players drafts are either hard-coded to match one of the maps utilized in our study, or they are drafted based on a drafting heuristic. The drafting phase occurs only once in the game. Following drafting, the agent executes the next three phases in sequence. First, in the "Reinforce" phase, the agent receives a specific number of reinforcements based on the number of territories and continents they control. The agent needs to assign the given reinforcements to the territories they control. Each country reinforced is an individual action. Next, the agent moves on to the "Attack" phase. In this phase, the agent can attack adjacent territories with their troops. Within each attack action, the agent specifies which opposing territory they would like to attack, along with the territory they would like to attack from. The agent must also specify the number of troops they would like to move into the opposing territory should the win the conflict. Each combat sequence between two territories is executed in a similar manner to the physical board game,

1. A maximum of three troops are chosen from the attacking territory, and a maximum of two troops are chosen from the defending territory

2. For both the attacker and defender, a number of die are rolled based on the number of troops involved in each attack.

3. The rolls are sorted in descending order, and each roll is compared between the attacking and defending country.

4. For each comparison, the country with the lower roll loses one troop. The defending territory wins all ties.

5. The above steps are repeated until either the attacking or defending player has been defeated.

Following combat, the agent can move all but one troop into the conquered territory. Once the agent has finished attacking, they move on to the final phase in their turn, "Freemove." In the "Freemove" phase, the player can move troops from one territory they control to another, as long as the territories are connected. Once the agent executes all their actions, the actions of the two agents are simulated and the player is reset to the "Reinforce" phase to start their next turn. The game is complete when either the agent is out of troops or controls all territories.

An action is specified by a four-item tuple, i.e. $< p, s, t, tr >$. The first item, $p$, specifies which type of action is being conducted, among the four possible phases in the game. Item two, $s$, denotes the source country for the action. For reinforce and drafting actions this is the country that the agent wants to add troops to, whereas for the attack and freemove actions, $s$ denotes the country you will be attacking or moving from. The, final two items, $t$ and $tr$, are specifically for attack and move actions. $t$ specifies the country that you would like to attack or move to. For the attack action, $tr$ specifies the number of troops you would like to move from the attacking country if you win the combat. When the agent specifies a move action, $tr$ denotes the number of troops to be moved from $s$ to $t$.

### I.1 State Space

The state of the game is stored as a dictionary. The state dictionary records information such as country ownership, number of troops on each country, continent ownership, etc. We also record information

about players such as number of reinforcements available to a player, number of players alive, current turn number, etc. We have provided six functions to encode the state space which can be passed as an input to a Reinforcement Learning model.

The first function encodes the state using 54 features. The initial 42 features contain country related information for each opponent (21 features each) and the next 5 features contain continent ownership data. The remaining features are used for other information related to the game like number of areas controlled by the player, troops left to be drafted by the player, troops left for reinforcement, number of players alive, current turn number and if the current turn belongs to the player.

The second function encodes the information in the form of one hots. It has a total of 132 features, the first 84 features contain information regarding country ownership as one hots, 21 each for the player, opponents and countries with no owner. The next 21 features denote the number of troops on each country. The next 20 features contain information regarding continent ownership, 5 each for the player, opponents and no owner. The remaining features contain other relevant information as described for the first function. For both of the first two functions described, we also provide normalized versions of these functions where all the real valued spaces are divided by a normalising constant.

The fifth encoding function contains all the 132 features of the third function and additional information for the current phase. It contains 134 features in total. This function returns normalised values. The last encoding function contains 298 features. The initial features are similar to the ones present in the third encoding function. Apart of that it explicitly contains information about where an agent or player can attack and execute a freemove. This information can help the reinforcement learning model more easily. This function also returns normalised values.

## I.2  Reward Functions

We have setup four different types of reward functions ranging from sparse to dense. The recommended reward function is the rules-based reward which provides rewards for successful actions, finishing a phase, successful action in a phase and winning the game. The rewards for winning the game are weighted by a factor of 10 compared to others which are weighted by a factor of 1.

The most simple reward function available is a sparse reward function which provides negative rewards for losing the game and positive rewards for winning the game. In order to increase the number of rewards given throughout the game, we created the turn count reward function which rewards the agent for every turn it plays. Survival reward function was built on top of this to provide an additional negative reward for losing apart from the reward for surviving.

## I.3  Human Drafting

Finally, we have also setup a functionality in our simulator that allows player or the opponents to skip the drafting phase and follow a fixed draft based on a predefined map. In such cases, we have predefined fifteen types of map initialisation containing troops for both opponents, which correspond to the exact maps utilized in our data collection procedure. Our setup chooses one of the map initializations and corresponding selections made by a participant in the user study to simulate the game.