# OpenReview forum: "A Computational Interface to Translate Strategic Intent from Unstructured Language in a Low-Data Setting"
_EMNLP/2023/Conference — EMNLP 2023 Findings_

### Official Review · Reviewer_Y69H · 2023-07-28

**Soundness:** 3

**Excitement:**

2: Mediocre: This paper makes marginal contributions (vs non-contemporaneous work), so I would rather not see it in the conference.

**Paper Topic And Main Contributions:**

The paper proposes a dataset of inferring constraints and goals from human descriptions, grounded in a broad game of Risk. The dataset has 1053 examples, and covers 6 predefined goals (each maps to a [-100, 100] human preference value) and 8 (or 9?) constraint types ("I must xxx (number) xxx").

Authors define a Roberta-based model with multiple classification heads, and fine it outperform a human baseline and a ChatGPT baseline.

**Questions For The Authors:**

- Table 5 lists 9 constraint types, but main paper says 8?

- What's the ChatGPT version (gpt-3.5-turbo?), and G.1 seems to use a very long zero-shot prompt, have you considered a shorter few-shot prompt that might work better?

**Reasons To Accept:**

- The problem of inferring (latent) human intentions is very important and interesting. The paper provides a good instantiation in a grounded game setup, and contributes a dataset that could be of interest to the NLP community.

- The design of multiple classification heads and the use of CE, MSE, OaXE losses are reasonable,

- The performance of the proposed model looks good, outperforming a human baseline and a ChatGPT prompting baseline.

**Reasons To Reject:**

- I'm not sure if the task is more like inferring or translation/semantic parsing, as the human input explicitly talks about these constraints and goals, and the main challenge is just to turn them into the structured format defined by the dataset, instead of any theory-of-mind or rational-speech-act kind of inferring. Also, in the intro, I'm not sure if "theory of mind" is just "shared understanding of each. team member of their role".

- Given the task seems easy, I don't get why Roberta model can beat human and ChatGPT. Some analysis or error breakdown will be very helpful, and my suspicion is just that Roberta model is specifically designed for the output format, which is not surprising when it outperforms other models. Also, why Table 1 chooses a different metric than other tables?

**Reproducibility:**

4: Could mostly reproduce the results, but there may be some variation because of sample variance or minor variations in their interpretation of the protocol or method.

**Reviewer Confidence:**

2: Willing to defend my evaluation, but it is fairly likely that I missed some details, didn't understand some central points, or can't be sure about the novelty of the work.

---

> ### Author Rebuttal · Authors · 2023-08-28
>
> We thank the reviewer for their review and for raising areas where we may clarify our contributions.
> - **Theory of Mind discussion** - We thank the reviewer for pointing out this concern regarding our usage of "theory of mind." The definition of theory of mind that we had in mind is the ability of the AI-agent to understand the mental state or desired behavior of a human specifier through commander's intent [Schaafsma et al 2015]. However, we do acknowledge that the broader concept of "theory-of-mind" is often bidirectional with regards to imparting mental states and will therefore remove it from the introduction. Our goal was to facilitate shared cognition between a human-AI team, specifically through aligning the values of the agent with the human’s needs. We will update the first paragraph of the introduction to more directly state this. We would also be happy to change the title of our task to “Automated Strategy Translation or Automated Strategy Parsing” rather than “Automated Strategy Inference”, if that is deemed by the reviewer to be a better fit.
> - **“my suspicion is just that Roberta model is specifically designed for the output format”** - Yes, one of the inherent advantages of the custom Roberta model is that it is designed for the specific output format, which is inherent in all custom models build to perform a task. ChatGPT also has inherent advantages which our Roberta model did not. For example, unlike our model, ChatGPT already had an abundance of knowledge about Risk Risk. We prompted ChatGPT to explain the rules of Risk and it provided us a very detailed response about the rules of risk, i.e.
>   - "Risk is a classic strategy board game that involves players trying to conquer territories and continents in order to achieve global domination. Here's how you play the game:
> Objective: The ultimate goal of Risk is to eliminate your opponents and achieve global domination by controlling all the territories on the game board.
> Setup:
> Game Board: Place the game board on a flat surface. The game board represents a world map divided into territories and continents.
> Armies: Each player selects a color and takes their corresponding army pieces (infantry, cavalry, and artillery) and a set of territory cards. The number of armies each player starts with depends on the number of players:
> 2 Players: 40 armies each
> 3 Players: 35 armies each
> 4 Players: 30 armies each
> 5 Players: 25 armies each
> 6 Players: 20 armies each
> Territories: Players place one army on each territory they control. Players take turns placing one army at a time until all territories are occupied........... (continued)"
>
>   Our model did not have this advantage and needed to learn about Risk from scratch. Each model comes with its inherent advantages and disadvantages which make it more or less suitable for a given task.
> - **“why Table 1 chooses a different metric than other tables”** - Table 1 depicts the raw scores, in terms of the average number of goals and constraints predicted correctly per example. Tables 2 and 3 report 10-fold cross-validation accuracies. Table 1 is depicted as such to more easily quantify the significant differences as per our statistical tests.
>
> Answering other specific questions posed by the reviewer:
> 1. **“Table 5 lists 9 constraint types, but main paper says 8?”** - We thank the reviewer for highlighting this typo. Table 5 is correct, there are 9 constraint classes. We will update this in the main paper.
> 2. **“What's the ChatGPT version”** - We used the default 3.5 version of ChatGPT. We will add this information to Section 5.2.
> 3. **“G.1 seems to use a very long zero-shot prompt, have you considered a shorter few-shot prompt that might work better”** - We used a one-shot chain-of-thought prompt which included an example with reasonings for each annotated goal and constraint. For this work, we wanted to provide ChatGPT the same information that we provided humans to ensure a fair comparison between both modalities. In future work, we plan to explore further prompt engineering approaches as well as the vertical of how many prompts are required for ChatGPT/other comparable LLMs to beat the performance of our custom Roberta model.

---

### Official Review · Reviewer_HrLg · 2023-08-06

**Soundness:** 4

**Excitement:**

3: Ambivalent: It has merits (e.g., it reports state-of-the-art results, the idea is nice), but there are key weaknesses (e.g., it describes incremental work), and it can significantly benefit from another round of revision. However, I won't object to accepting it if my co-reviewers champion it.

**Paper Topic And Main Contributions:**

This works study translating natural language descriptions to goals and constraints in a strategic game called RISK. This work starts from data curation to form the first benchmark on mapping language description + state (troop selection) to a set of constraints and goals. Then, they propose augmenting data through paraphrasing. This work also designs a goal extraction and constraint extraction model with Roberta. The experiment demonstrates that the small model proposed in this work achieves more accurate goal and constraint prediction compared to humans and ChatGPT.

**Questions For The Authors:**

* How familiar are the annotators with RISK (S 3.3)?
* How familiar are the human participants with RISK (S 5.1)?

**Reasons To Accept:**

* This work is comprehensive, it covers the full pipeline from data collection to the evaluation of different approaches
* The experiment design is solid. It compares the proposed method with both human participants and strong LLMs such as ChatGPT
* Converting the language descriptions to a set of goals and constraints is well-grounded in the literature, and the curated dataset in this work covers many aspects which are missing in the previous works.

**Reasons To Reject:**

* The quality of the annotation is somewhat questionable. It is unclear what level of expertise the annotators have on RISK. According to Section 3.3, the annotators seem to be hired randomly. There is only one annotation per example. By looking at the annotated example shown in Figure 1, I also found the example is somewhat **subjective** (This is perhaps due to I don't have access to the full annotation instruction.) However, the above-mentioned issues worry me about the annotation quality -- can we really use the annotation as the ground truth for experiments? The proposed model trained with the augmented data seems to perform the best, however, if the annotation quality is questionable, the better performance can be due to overfitting to annotation artifacts (e.g., one annotator's personal preference). The lower human performance in S5.1 somewhat reflects this concern. It will be helpful to discuss the annotation quality.
* How useful translating language description to goals and constraints is unclear. The goals and constraints ultimately serve as the **intermediate** stages of the game playing. It is unclear how much improvement a system's winning rate will have with `X%` gain from better goal and constraint extraction. Such experiments often require an end-to-end system and I understand it may go beyond the scope of this work. It will be helpful to have a quantitative study. For instance, the work may design experiments with human players by asking about their preferences on different models' output during *real* games.

**Reproducibility:**

3: Could reproduce the results with some difficulty. The settings of parameters are underspecified or subjectively determined; the training/evaluation data are not widely available.

**Reviewer Confidence:**

4: Quite sure. I tried to check the important points carefully. It's unlikely, though conceivable, that I missed something that should affect my ratings.

**Typos Grammar Style And Presentation Improvements:**

* The color text in the upper-left box of Figure 1 is confusing. It took me a while to understand it aligns with contents in constraints and goals boxes other than troops. It may be helpful to have an explanation
* The introduction of "Commander's intent" comes too suddenly. Maybe also explain why studying "Commander's intent" is interesting/beneficial

---

> ### Author Rebuttal · Authors · 2023-08-28
>
> We thank this reviewer for their time and thorough review of our submission. We hope to address each individual point from the reviewer through the following:
> - **“The quality of the annotation is somewhat questionable”** - Yes, as per any crowdsourced dataset, our annotations will be noisy. However, we have employed various measures to ensure data integrity. Firstly, we utilized the master's qualification filter on Mechanical Turk. Secondly, each datapoint was individually reviewed and filtered by two researchers to ensure that the goals and constraints were representative of the language descriptions. Third, we bucketted goal values into 5 uniform buckets (from highly irrelevant to highly relevant) instead of using the original, raw values in [-100,100]  (lines 356 - 360), which we found important to address user-specific variation in ratings.
> - **“the annotators seem to be hired randomly”** - Yes, annotators were hired randomly as is standard practice in online human-subjects research. We did not want to restrict our data to any specific population of users in order to bias our data. However, while the users were not guaranteed to be Risk experts, we provided an extensive tutorial which explained the rules of the game. Crucially, participants also needed to pass a 5-question quiz in order to proceed to the study (lines 285 - 290), thereby confirming that they have gained the necessary expertise in order to complete the task. We will include the full quiz in the appendix in the camera-ready version.
>  - **“The goals and constraints ultimately serve as the intermediate stages of the game playing”** - Goals and constraints serve as a strategy to guide the policy of an agent in order to adhere to the specifications of a human. We are not specifically interested in getting players to win the game of Risk or understanding how much more likely you are to win the game of risk through the use of strategy specifications. Our goal is to facilitate strategy inference with a view towards real-world human-AI teaming applications wherein the human requires their AI teammate to perform the task in a specific way which conforms with the goals of the team/commander. Risk was an adequate proof-of-concept domain as it shares many similar properties (resource scheduling, troop allocation, multi-step planning, etc.) to real-world human-AI teaming problems such as search and rescue or drone scheduling. In future work, we hope to expand to these domains and also expand our approach by developing a Seldonian optimization method to allow an RL-agent to adapt its learned policy by utilizing the goals and constraints inferred through a user's language strategy.
>
> Here are our responses to the specific questions posed by the reviewer:
> 1. **“How familiar are the annotators with RISK (S 3.3)?”** - We did not specifically recruit risk experts. We taught users the rules of the game through a tutorial and assessed minimum performance via a quiz. More details have been provided in our earlier answer under the “annotators seemed to be hired randomly” section.
> 2. **“How familiar are the human participants with RISK (S 5.1)?”** - We did not specifically recruit participants with experience in playing Risk. We did however provide them with a tutorial as well as visually annotated examples to familiarize them with the domain and task. Our human-evaluation task also included attention checks in each annotation to ensure that participants were providing authentic annotations.
>
> Finally, here are some addition clarifications to the minor comments:
> - **“The color text in the upper-left box of Figure 1 is confusing”** - We regret the confusing color scheme here. The green text in the language description corresponds to the text relating to the constraints, and the orange text corresponds to text related to goals. The blue text is generic text in the description that does not directly relate to any goal or constraint. We will change that to black to avoid any confusion as well as add a description of the color scheme to the caption.
> - **“The introduction of "Commander's intent" comes too suddenly”** - Commander's Intent has been a widely used formulation for engendering an understanding of a mission or a plan for a team. The point of specifying Commander's Intent is for each team member to know what individual actions should be taken to accomplish a mission while still adhering to the goals and constraints of the commander. Commander’s Intent allows a commander to state what they value and design a reward structure around those values. Commander's Intent could similarly function as a helpful scaffold for a human commander or teammate to provide their intent to an autonomous agent. In instances where an agent needs to execute a plan or a mission independent of the human user, the human needs a mechanism for providing their intent such that the autonomous agent can identify the specific set of low-level actions that adhere to the goals and constraints specified. In the camera-ready version of this paper, we will expand on this description of Commander's Intent and why it is relevant

---

### Official Review · Reviewer_XhaB · 2023-08-11

**Soundness:** 3

**Excitement:**

3: Ambivalent: It has merits (e.g., it reports state-of-the-art results, the idea is nice), but there are key weaknesses (e.g., it describes incremental work), and it can significantly benefit from another round of revision. However, I won't object to accepting it if my co-reviewers champion it.

**Paper Topic And Main Contributions:**

The paper proposes a new task/paradigm - that of identifying strategic intent from highly unstructured language - in the setting of a war-themed board game. For the game in question (Risk), a number of goals and constraints are defined, and a dataset of unstructured plans mapping to goals and constraints is collected using crowdworkers. Then, a RoBERTa model is finetuned to identify these goals and constraints. Results show improvement over prompting ChatGPT.

**Reasons To Accept:**

The proposed task is interesting, and the results are fairly insightful in showing that models can do better than humans on this reasonably complex task.

**Reasons To Reject:**

It is hard to gauge the applicability of the proposed framework to tasks not following the exact paradigm that Risk falls under. In other words, it may not be easy to find games, or other applications, where the proposed approach would work seamlessly. A new dataset would, clearly, need to be collected for any such effort.

The ChatGPT comparison may be unfair, since (to the best of my understanding) only one input example is provided to ChatGPT, which differs significantly from the training data the RoBERTa model gets.

**Reproducibility:**

3: Could reproduce the results with some difficulty. The settings of parameters are underspecified or subjectively determined; the training/evaluation data are not widely available.

**Reviewer Confidence:**

2: Willing to defend my evaluation, but it is fairly likely that I missed some details, didn't understand some central points, or can't be sure about the novelty of the work.

---

> ### Author Rebuttal · Authors · 2023-08-28
>
> We thank the reviewer for their time in reading and reviewing our paper. We hope to individually address each point through the following:
> - **“A new dataset would, clearly, need to be collected for any such effort”** - Yes, as stated by this reviewer, any new domain would require a new dataset. However, our task of automated strategy inference is inherently domain specific, in that there are many implicit details of the domain that need to be understood by the model in order to adequately perform the task. In many other domain specific tasks, the ML-practitioner often requires a small dataset to personalize or adapt to a new domain or user (for example, Li et al 2022, Morishita et al 2022, from the EMNLP 2022 proceedings). By providing an end-to-end pipeline for automated strategy specification, we have provided all the tools necessary (i.e, data collection/augmentation methodology, ML architecture, evaluation)  for future work to build on this work in other domains in a low-data setting (1000 data points).
> - **“it may not be easy to find games, or other applications, where the proposed approach would work seamlessly”** - We thank this reviewer for highlighting the importance of domain transfer. Firstly, there are many games/domains which require similar goals and constraints to our chosen domain, Risk. These games include Starcraft, Age-of-Empires, Dota 2, etc, which all have similar gameplay properties relating to troop allocation and resource scheduling. Furthermore, there are many applications in the domains of robotics and embodied agents (i.e. domains relating to navigation and teaming) which utilize formal planning specifications, such as goals and constraints. Some examples include search and rescue drones, which need to deploy policies specific to any given situation, or robot scheduling, which require humans to deploy robots with a plan to schedule activities on a factory floor. The scaffold of goals and constraints would be highly suitable in such instances.
> - **“The ChatGPT comparison may be unfair”** - Firstly, as stated in lines 512 - 515, our ChatGPT prompt included one annotated example as we wanted to provide the same information to ChatGPT as we provided to humans during the human-evaluation. Therefore, it enabled us to compute a fair comparison between humans and ChatGPT. Furthermore, our prompt was designed to follow the widely used chain-of-thought prompting approach [Wei et al. 2023], wherein, in our prompt we include the chain-of-thought reasoning for the annotation of each goal and constraint within the example (see Appendix G). Unlike our model, ChatGPT also had an inherent understanding of Risk gameplay and strategy games, whereas our Roberta model needed to learn to infer this information from scratch. For these reasons, we believe that we have provided ChatGPT with enough information to perform the task and facilitate a fair comparison with our other baselines.

---

### Official Review · Reviewer_ASwe · 2023-08-11

**Soundness:** 3

**Excitement:**

4: Strong: This paper deepens the understanding of some phenomenon or lowers the barriers to an existing research direction.

**Paper Topic And Main Contributions:**

The paper presents a computational interface that automatically infers strategic intent from unstructured language. In many real-world scenarios, humans and AI collaborate, requiring the AI to understand and act upon the strategic intent specified by humans. The paper introduces a model that translates unstructured language into machine-readable goals and constraints. This approach is demonstrated in a game environment, where the model learns to derive intent from language-based game strategies. The results show that the proposed model outperforms human interpreters and ChatGPT in interpreting strategic intent from language, especially in low-data scenarios.

**Questions For The Authors:**

Could the proposed model achieve multi-round inference? In practice, many of the strategic games are sequential decisions.

**Reasons To Accept:**

1. The paper is well presented.
2. The authors collected 1053 samples for RISK game on Amazon Mechanical Turk, which is extensive.
3. The research showcases empirical results, proving the efficacy of the proposed approach against human interpreters and ChatGPT, providing a clearer understanding of the model's capabilities.

**Reasons To Reject:**

1. The participants are asked to provide natural language descriptions. In practice, people may want to use language together with some symbolic information and mathematical specifications. It will be beneficial if he model could handle a mixture of different modalities.
2. The collected data are only one step with randomized configurations. RISK is a multi-round game. The player may adapt their strategies and inference methods according to other players. Making the original multi-round game as a single step one simplifies it too much. This is my main concern.

**Reproducibility:**

4: Could mostly reproduce the results, but there may be some variation because of sample variance or minor variations in their interpretation of the protocol or method.

**Reviewer Confidence:**

3: Pretty sure, but there's a chance I missed something. Although I have a good feel for this area in general, I did not carefully check the paper's details, e.g., the math, experimental design, or novelty.

---

> ### Author Rebuttal · Authors · 2023-08-28
>
> We thank the reviewer for their measured review and for raising areas wherein we may clarify our contribution. Through the following response, we hope to clarify each question and critique raised in the review.
>  - **“It will be beneficial if the model could handle a mixture of different modalities”** - We agree that multimodality will be beneficial. In this paper, we develop a solution to a novel task: our goal was to first establish a language approach that can help end-users interface with planning tools. Furthermore, our approach included a second modality: the user’s suggested troop selections. While multimodality is a highly valuable contribution towards our “Automated Strategy Inference task, a multimodal strategy translation module would constitute a novel contribution in its own paper. In our camera-ready version, we will add a discussion of the potential to incorporate additional modalities and the need for modalities such as symbolic information and mathematical specifications.
> - **“Could the proposed model achieve multi-round inference?”** - We thank the reviewer for this question. In our paper, we developed an approach to distill assignments for goals and constraints from users’ unstructured language descriptions. We specifically chose the draft phase as the draft phase is the primary phase which requires the users to develop and deploy their strategy and was thus the most suitable phase for our intervention. It is true that as the game proceeds, the user may want to provide modifications to their strategy. Instead of allowing the user to specify a new strategy to account for these modifications, a more appropriate approach could be to employ complementary methodologies similar to prior work wherein, language inputs/commands are utilized to update or change the sub-goals the agent is considering [Fu et al. 2019,  Goyal et al. 2019]. Such approaches could be utilized to individually update specific goals or constraints over the course of multiple rounds of the game to adapt an initially specified strategy, for example, “I am no longer interested in keeping my troops close together” would reduce the value for the specified goal. Recent work has also shown promising results with regards to ChatGPT’s zero-shot performance for dialog state tracking (Heck et al 2023). Therefore, we hope to experiment with deploying ChatGPT, or other SOTA LLMs, in a task-oriented dialog setup, to converse with a user to update the initially specified strategy after multiple rounds of the game, however, this is currently outside the scope of this work and remains a contribution for future work.

---

### Meta-Review · Area_Chair_dXHL · 2023-09-18

**Recommendation:** 4

**Metareview:**

This paper introduces a model that translates unstructured natural language into structured goals/constraints allowing the “intent” or meaning of natural language to be converted into logical representations tied to the underlying world state. Their pipeline shows how such data can be collected and then augmented allowing better training of a model on such data. They demonstrate improved performance of this explicit goal/constraint decoder over a baseline that predicts outputs sequentially with in-context learning as well as a human baseline and show improved performance over both. Most reviewers agree that this is an interesting paper and the dataset/pipeline creation and improved performance of the model are good contributions to the language/RL community. I urge the authors to consider the suggestions of better visualisation/examples in the paper and quality checks of the datasets, but even it's current state the methodology presented here is a valuable contribution to future work in language/RL/game environments.

---

### Decision · Program_Chairs · 2023-10-07

**Decision:**

Accept-Findings

**Comment:**

This paper introduces a model that translates unstructured natural language into structured goals/constraints allowing the “intent” or meaning of natural language to be converted into logical representations tied to the underlying world state. Their pipeline shows how such data can be collected and then augmented allowing better training of a model on such data. They demonstrate improved performance of this explicit goal/constraint decoder over a baseline that predicts outputs sequentially with in-context learning as well as a human baseline and show improved performance over both. Most reviewers agree that this is an interesting paper and the dataset/pipeline creation and improved performance of the model are good contributions to the language/RL community. I urge the authors to consider the suggestions of better visualisation/examples in the paper and quality checks of the datasets, but even it's current state the methodology presented here is a valuable contribution to future work in language/RL/game environments.